# Time-Series Forecasting of Seasonal Data Using Machine Learning Methods

**Vadim Kramar** * **and Vasiliy Alchakov** 

Department of Informatics and Control in Technical Systems, Sevastopol State University,
299053 Sevastopol, Russia
* Correspondence: kramarv@mail.ru

**Abstract:** The models for forecasting time series with seasonal variability can be used to build automatic real-time control systems. For example, predicting the water flowing in a wastewater treatment plant can be used to calculate the optimal electricity consumption. The article describes a performance analysis of various machine learning methods (SARIMA, Holt-Winters Exponential Smoothing, ETS, Facebook Prophet, XGBoost, and Long Short-Term Memory) and data-preprocessing algorithms implemented in Python. The general methodology of model building and the requirements of the input data sets are described. All models use actual data from sensors of the monitoring system. The novelty of this work is in an approach that allows using limited history data sets to obtain predictions with reasonable accuracy. The implemented algorithms made it possible to achieve an R-Squared accuracy of more than 0.95. The forecasting calculation time is minimized, which can be used to run the algorithm in real-time control and embedded systems.

**Keywords:** seasonal time series; machine learning; forecasting; XGBoost; Prophet; SARIMA; Long Short-Term Memory; Holt-Winters Exponential Smoothing



## 1. Introduction

Generally, a time series is a set of data ordered in time. Each time series element is assigned a unique Timestamp ($TS$) index. The distance between indexes can be seconds, hours, days, months, or even years. At that time, the value of the time interval between neighboring indexes is usually assumed to be the same and is denoted by $T_s = TS[k] - TS[k-1]$. For example, the change in quotes on the exchange can be described by the set of real numbers $Y[k]$ with an interval with $T_s = 1$ min, and the quotes are accumulated in the data set from the start of trading on the exchange until its closure. Some finite sample $Y$ is considered at a specific time interval $t \in \left[t_0, t_0 + T_s, \ldots, t_f\right]$. In general, a time series can be represented by a function of the form $Y_t = T_t + S_t + R_t$, where $T_t$ is the trend cycle component (Trend), $S_t$ is the seasonal component (Seasonal), and $R_t$ are the residuals (Residuals). Time series containing a seasonal component $S_t$ has periodicity properties. That is, the character of the variability of the time series repeats from time to time. An example is the growth of purchasing power during the holiday season or the growth of passenger traffic during the vacation season. Modern machine learning methods can create predictive models for this type of time series, which allows calculating the estimate (forecasting) of the parameter for several steps forward relative to the current point in time (forecast horizon) [1–6]. In this case, to train the model, we need only knowledge of the history of the parameter, which is described by the time series, that is, the historical data on the time interval, the length of which corresponds to several periods of seasonal variability $T$. The knowledge of other parameters, as a rule, is not required. Thus, the task arises of building models which can estimate some experimental parameters with seasonal variability on some forecast horizon based on information about previous and present values of this parameter (historical data). Currently, the analysis and forecasting of time series

with seasonal variability are most common in financial markets and trading [7–10]. The works [11–14] build models to obtain forecasts of commodity prices and rates of cryptocurrencies. The works [15,16] forecast the value of tourist flow, depending on the time of year. In turn, tourist flows impact changes in passenger traffic by various modes of transport, as shown in [17,18]. Processes that affect climate [19,20], ecology [21,22], and medicine [23,24] are also subject to seasonal changes. The authors [25–28] describe the application of time series with seasonal variability in industry and energy and resource consumption. The application of deep learning technology and long short-term memory network for physical systems is shown in [29,30]. All research above uses practically the same set of methods and algorithms. First, these are methods for obtaining autoregressive models ARIMA (Autoregressive Integrated Moving Average) and SARIMA (Seasonal ARIMA), which have been known for quite a long time, and the mathematical apparatus of which is well developed. The first mention of the method was made in 1970 by George Box and Gwilym Jenkins. ARIMA/SARIMA allows us to obtain the primary class of models used for time series analysis and forecasting. The Holt-Winters Exponential Smoothing method, developed by Charles Holt in the late 1950s, and its modern modification, the Exponential Smoothing Algorithm (ETS), are also used. In addition to the above methods, which can already be classified as classical, new algorithms have appeared in recent years, successfully coping with constructing predictive models of time series with seasonal variability. These are the methods of Facebook Prophet [31], XGboost [32], and a set of Artificial Neural Network (ANN) methods, in particular, the Long Term Memory (LSTM) method [33–35].

This article presents the results of applying the above methods to build a model to predict the inlet flow of wastewater from a wastewater treatment plant. This model can be used to calculate the optimal loading of equipment and to ensure a given quality of treatment, which in turn will significantly reduce the consumption of electrical energy by the plant. The methodology of data preparation and the specifics of applying each of the methods are described, the statistical analysis of the obtained results is performed, and recommendations for improving the forecast quality are given. All numerical calculations and implementation of algorithms were performed using the Python language and specialized libraries [36]. The novelty of the work lies in the use of machine learning methods for predicting the technological process. The forecast results are used in the real-time control system. A method for preparing data for use in predictive models is proposed. A feature of the approach is using data over a short time interval. A sample for the last 15–17 days is used for training. In this case, the forecast is made for the next two days. Automatic retraining of the model is used every 24 h to maintain accuracy of the model. At the same time, it takes less than 10 s to train a new model, and the accuracy of the model remains unchanged. As a result, several methods were chosen that optimally solve the problem. The developed algorithms can be easily integrated into embedded control systems or software tools for real-time treatment plant control and do not require significant computing resources.

The Section 2 describes the main approaches used for data preprocessing. The data format is defined, and a technical map is proposed for solving the problem. The Section 3 discusses the main provisions of the methods and provides recommendations for using Python libraries. The graphical results of checking each model on a test data set are also given. For the XGBoost and LTSM methods, an algorithm for generating a synthetic data set for training is given, which must be obtained from the original data set to build a prediction model. The Section 4 describes the statistical metrics used to compare the accuracy of the models. The results of the numerical calculation of the metrics are given, as well as the time spent on the preparation and use of each model. The Section 5 summarizes the findings and outlines areas for further research.

## 2. Methodology

The study aimed to find, using known approaches, the optimal algorithm that would allow building a short-term forecast in the shortest time and with the least initial dataset (historical data).

The main idea and novelty of the implemented approach are that our algorithm does not use the same machine-learning model. The model is built again using new data every 24 h. That is why short-term samples of initial data are used for training and testing models. This idea allows us to consider new factors that cannot be included in the long-term model, such as equipment replacement, maintenance, and unplanned load changes. It keeps the forecast accuracy at a high level, as well.

Below is a methodology that allows us to implement the described approach effectively.

### 2.1. The Technical Roadmap and Data Collection

In this research, we used the actual data obtained from the monitoring system of the wastewater treatment plant. The monitoring system sensors collect basic parameters, such as input and output flow value, oxygen concentration, ammonia content, etc. Data from sensors come to the OPC server, and the OPC monitor records data with a frequency of once per second in synchronous and asynchronous modes. After the preprocessing, the monitoring system saves data in the database. The monitoring system operates 24/7, allowing obtaining data for any period of interest in the form of a comma-separated values format.

The model development process can generally be presented as a technical roadmap, shown in Figure 1.

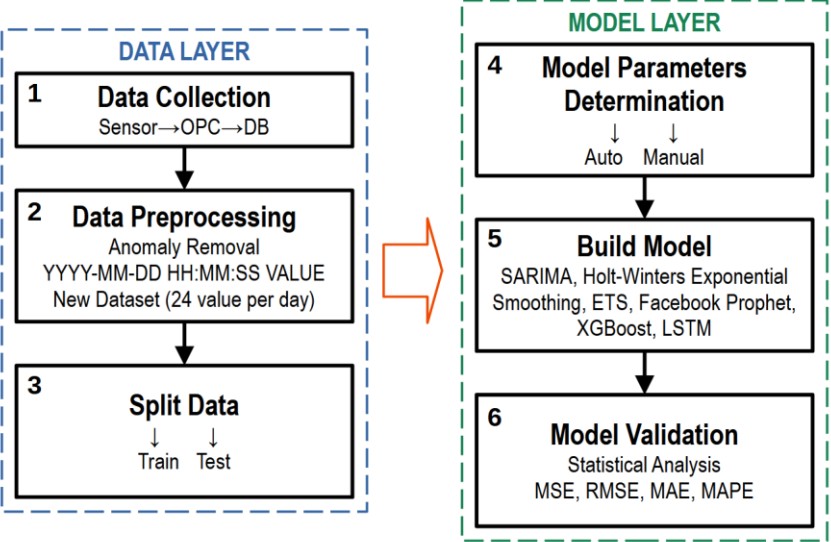

**Figure 1.** The technical roadmap of data analysis.

There are two layers: the data layer and the model layer. Each layer is conventionally divided into three steps.

As described above, the data layer (steps 1–3) implements preprocessing and data preparation.

The model layer involves selecting model parameters (step 4), which can be implemented in two ways: automatically and manually. The selection method depends on the library's capabilities to build the model. The next step (step 5) is model building, which involves setting the parameters defined in the previous step and training the model. At this step, all actions with the model can be called «model building». The last step (step 6) is the validation of the model. By «model validation», we mean obtaining a prediction and comparing the parameter estimates with their actual values. To assess the quality of the model, we will use a set of metrics described below. If the model metrics meet the specified ones, the model is saved and can be used to generate a forecast. If the metrics are unsatisfactory, you should return to step 4 and perform all subsequent steps at the data layer.

The estimation of the model's accuracy was calculated with five statistical metrics: correlation coefficient ($R^2$), mean squared error ($MSE$), root mean squared error ($RMSE$),

mean absolute error (*MAE*), and mean absolute percentage error (*MAPE*) [12,37]. The correlation coefficient ($R^2$) is used to determine the degree of fit of the forecasted values to the actual values. *MSE* is the average change in actual and forecast values, *RMSE* is the square root of *MSE*, and *MAE* is proposed as the average change between actual and predicted values. Compared with *MAE*, *RMSE* emphasizes the variance between data outliers. These metrics are calculated with equations:

$$
\begin{aligned}
R^2 &= 1 - \frac{\sum_{i=1}^{m} |\hat{y}_i - y_i|^2}{\sum_{i=1}^{m} \left| \frac{1}{m} \sum_{i=1}^{m} y_i - y_i \right|^2} \\
MSE &= \frac{1}{m} \sum_{i=1}^{m} |\hat{y}_i - y_i|^2 \\
RMSE &= \sqrt{\frac{1}{m} \sum_{i=1}^{m} |\hat{y}_i - y_i|^2} \\
MAE &= \frac{1}{m} \sum_{i=1}^{m} |\hat{y}_i - y_i| \\
MAPE &= 100\% \frac{1}{m} \sum_{i=1}^{m} \frac{|\hat{y}_i - y_i|}{|y_i|}
\end{aligned}
\tag{1}
$$

where $y_i$ and $\hat{y}_i$ are the actual and predicted values, and *m* represents the number of samples.

*2.2. Data Preprocessing*

The data from the monitoring system sensors are stored in the database in raw form. This means that a simple upload of a parameter of interest over a specific time interval cannot be used as an input data set for model training. Pre-processing and filtering are required. The data format to be used in the vast majority of methods can be represented in general terms as $DD - MM - YYYY\ HH : MM : SS\ VALUE$, where $YYYY$ is a four-digit year code, $MM$ is a two-digit current month code, $DD$ is a two-digit day code, $HH$ is a two-digit hour code, $MM$ is a two-digit minute code, $SS$ is a two-digit seconds code, and $VALUE$ is the real number corresponding to the parameter value obtained from the sensor.

As was mentioned above, the parameters are recorded with a frequency of 1 time per second. Since the resources of the monitoring system are limited, data averaged over 1 min are recorded in the database for each observed parameter. This frequency is also redundant for building an input flow prediction model, so the raw data obtained from the database underwent preprocessing. First, the data containing obvious outliers and anomalous values were discarded. At the next processing stage, the data were averaged over 1 h (resampling). As a result, a sample of data was generated for 19 days of observations, in which each day was represented by a set of 24 values of the observed parameter. The data set corresponding to the first 17 days of observations was used as training data. The remaining data were used to test the quality of the models. All preprocessing was performed using Pandas (Python Data Analysis Library) [38].

Table 1 shows a data slice of the entire dataset (total of 457 rows) obtained after applying the preprocessing procedure.

Figure 2 shows the periodicity of the process, with a seasoning period of 24 h. The data in the green area (Test Data) have been used to test the model quality, and the data to the left of the green area (Train Data) have been used to train the model. The task is formulated as follows: to build a predictive model using the available training dataset and use the model to build a forecast of the input stream for the next 48 h. Further steps are reduced to constructing and validating the obtained models using machine learning methods, discussed at the article's beginning.

**Table 1.** The slice of the entire dataset.

| ID | Timestamp | Measurement |
|---|---|---|
| 0 | 12 December 2022 00:00:00 | 6.305967 |
| 1 | 12 December 2022 01:00:00 | 5.355895 |
| 2 | 12 December 2022 02:00:00 | 4.122726 |
| 3 | 12 December 2022 03:00:00 | 3.546737 |
| . . . | | |
| 452 | 30 December 2022 20:00:00 | 6.578148 |
| 453 | 30 December 2022 21:00:00 | 6.591513 |
| 454 | 30 December 2022 22:00:00 | 6.785699 |
| 455 | 30 December 2022 23:00:00 | 6.396174 |
| 456 | 31 December 2022 00:00:00 | 6.322228 |

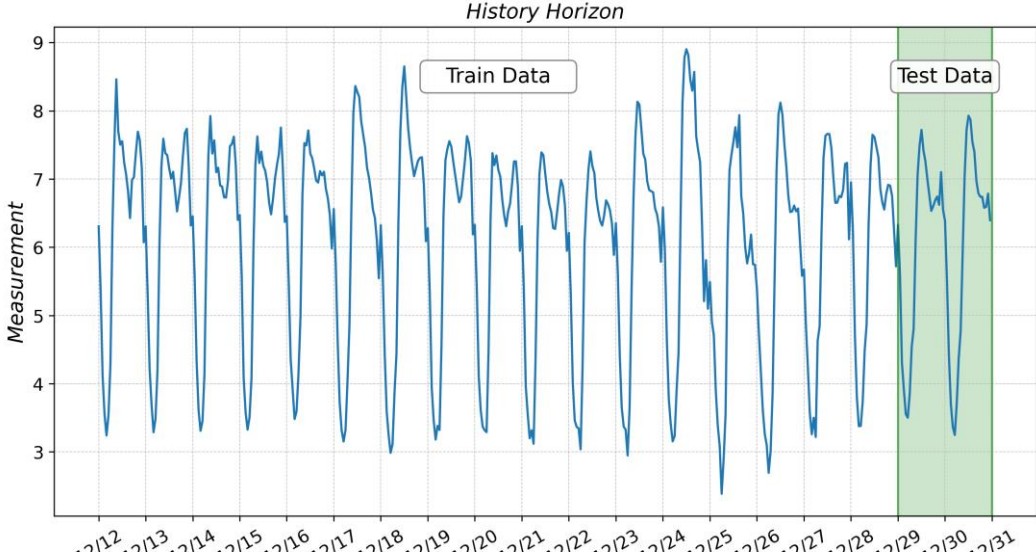

**Figure 2.** Initial dataset with a breakdown into training and test samples.

The following section will outline the main theoretical provisions and peculiarities of implementing the above methods using Python and open-source libraries.

## 3. Development of the Models

### 3.1. SARIMA

Seasonal Autoregressive Integrated Moving Average (SARIMA), or Seasonal ARIMA, is an extension of ARIMA that explicitly supports univariate time series data with a seasonal component. According to [39], the seasonal ARIMA model includes autoregressive and moving average terms at lag $s$.

The seasonal $ARIMA(p,d,q)(P,D,Q)_s$ model can be most succinctly expressed using the backward shift operator.

$$\Theta_P(B^s)\theta_p(B)(1-B^s)^D(1-B)^d x_i = \Phi_Q(B^s)\phi_q(B)\omega_t. \tag{2}$$

In Equation (2), $\Theta_P$, $\theta_p$, $\Phi_Q$ and $\phi_q$ are polynomials of orders $P$, $p$, $Q$ and $q$, respectively. In general, the model is non-stationary, although, if $D = d = 0$ and the roots of the characteristic equation (polynomial terms on the left-hand side of Equation (1)) all exceed unity in absolute value, the resulting model would be stationary.

Thus, it is necessary to check the original dataset for stationarity before using the method. This can be performed using the Augmented Dickey-Fuller (ADfuller) test. This test is based on a hypothesis, where, if the $p$-value is less than 0.05, then we can consider the time series stationary, and if the $p$-value is greater than 0.05, then the time series is non-stationary.

To perform the ADfuller test and implement the SARIMA method, the library statmodels is used [40]. ADfuller test result are as follows:

```
Augmented Dickey-Fuller Test for SARIMA method
1.  ADF : −6.162488723252326
2.  P-Value :  7.118446442706881e−08
3.  Num Of Lags :  18
4.  Num Of Observations Used For ADF: 389
5.  Critical Values :
        1% :  −3.447272819026727
        5% :  −2.868998737588248
       10% :  −2.5707433189709294
```

Thus, the original time series is stationary.

The selection of parameters $(p, d, q)(P, D, Q)$ can be performed automatically using the *auto_arima* function of the pmdarima library [41]. The result of the function is:

```
AutoARIMA. Performing stepwise search to minimize aic
ARIMA(2,0,1)(0,1,2)[24] intercept :  AIC=191.369, Time=44.95 sec
ARIMA(1,0,2)(0,1,2)[24] intercept :  AIC=184.365, Time=28.39 sec
ARIMA(1,0,2)(0,1,1)[24] intercept :  AIC=202.174, Time=3.42 sec
ARIMA(1,0,2)(1,1,2)[24] intercept :  AIC=inf, Time=31.94 sec
ARIMA(1,0,2)(1,1,1)[24] intercept :  AIC=inf, Time=5.80 sec
ARIMA(0,0,2)(0,1,2)[24] intercept :  AIC=218.241, Time=13.98 sec
ARIMA(2,0,2)(0,1,2)[24] intercept :  AIC=177.020, Time=39.47 sec
ARIMA(2,0,2)(0,1,1)[24] intercept :  AIC=194.216, Time=5.32 sec
ARIMA(2,0,2)(1,1,2)[24] intercept :  AIC=inf, Time=51.64 sec
ARIMA(2,0,2)(1,1,1)[24] intercept :  AIC=inf, Time=9.51 sec
ARIMA(3,0,2)(0,1,2)[24] intercept :  AIC=176.735, Time=46.17 sec
ARIMA(3,0,2)(0,1,1)[24] intercept :  AIC=191.289, Time=8.32 sec
ARIMA(3,0,2)(1,1,2)[24] intercept :  AIC=inf, Time=54.78 sec
ARIMA(3,0,2)(1,1,1)[24] intercept :  AIC=inf, Time=9.91 sec
ARIMA(3,0,1)(0,1,2)[24] intercept :  AIC=177.137, Time=37.80 sec
ARIMA(3,0,3)(0,1,2)[24] intercept :  AIC=181.003, Time=48.34 sec
ARIMA(2,0,3)(0,1,2)[24] intercept :  AIC=177.237, Time=51.24 sec
ARIMA(3,0,2)(0,1,2)[24]            :  AIC=169.441, Time=44.18 sec
ARIMA(3,0,2)(0,1,1)[24]            :  AIC=185.525, Time=9.49 sec
ARIMA(3,0,2)(1,1,2)[24]            :  AIC=inf, Time=52.88 sec
ARIMA(3,0,2)(1,1,1)[24]            :  AIC=inf, Time=9.48 sec
ARIMA(2,0,2)(0,1,2)[24]            :  AIC=178.763, Time=26.75 sec
ARIMA(3,0,1)(0,1,2)[24]            :  AIC=178.903, Time=25.43 sec
ARIMA(3,0,3)(0,1,2)[24]            :  AIC=182.748, Time=31.08 sec
ARIMA(2,0,1)(0,1,2)[24]            :  AIC=177.159, Time=28.53 sec
ARIMA(2,0,3)(0,1,2)[24]            :  AIC=178.965, Time=39.95 sec

Best model:  ARIMA(3,0,2)(0,1,2)[24]
Total fit time:  865.904 seconds
```

Where $s = 24$ is the period of the seasonal time series in hours.

After the model is built considering the obtained optimal parameters, in order to make a prediction, it is sufficient to use the *get_forecast* method of the model object, passing as a parameter the number of time counts, for which we need to obtain a prediction. Since the interval between the samples in the initial dataset is 1 h, and the forecast is made for 48 h, you should specify the input parameter steps = 48.

Figure A1 (see Appendix A) shows the time series plot obtained with the SARIMA model (solid blue line on the plot) and the actual measurement values from the test data set (dark bullet marker on the plot). The Section 4 will give a quantitative assessment of the quality of this and subsequent predictive models.

### *3.2. Holt-Winters Exponential Smoothing*

The Holt-Winters seasonal method comprises the forecast equation and three smoothing equations—one for the level $\ell_t$, one for the trend $b_t$, and one for the seasonal component $s_t$, with corresponding smoothing parameters $\alpha$, $\beta^*$ and $\gamma$.

The parameter $m$ is used to denote the seasonality period [42]. Two variations of this method differ from the seasonal component—the additive and multiplicative methods.

The component form for the additive method is:

$$\begin{aligned}
\hat{y}_{t+h|t} &= \ell_t + hb_t + s_{t+h-m(k+1)} \\
\ell_t &= \alpha(y_t - s_{t-m}) + (1-\alpha)(\ell_{t-1} + b_{t-1}) \\
b_t &= \beta^*(\ell_t - \ell_{t-1}) + (1-\beta^*)b_{t-1} \\
s_t &= \gamma(y_t - \ell_{t-1} - b_{t-1}) + (1-\gamma)s_{t-m} \, .
\end{aligned} \tag{3}$$

The component form for the multiplicative method is:

$$\begin{aligned}
\hat{y}_{t+h|t} &= \ell_t + hb_t + s_{t+h-m(k+1)} \\
\ell_t &= \alpha(y_t - s_{t-m}) + (1-\alpha)(\ell_{t-1} + b_{t-1}) \\
b_t &= \beta^*(\ell_t - \ell_{t-1}) + (1-\beta^*)b_{t-1} \\
s_t &= \gamma(y_t - \ell_{t-1} - b_{t-1}) + (1-\gamma)s_{t-m} \, .
\end{aligned} \tag{4}$$

The $k$ in Equations (3) and (4) is the integer part of $(h-1)/m$. The level equation $\ell_t$ shows a weighted average between the seasonally adjusted observation $(y_t - s_{t-m})$ and the non-seasonal forecast $(\ell_t - \ell_{t-1})$ for time $t$. The trend equation $b_t$ is identical to Holt's linear method. The seasonal equation shows a weighted average between the current seasonal index, $(y_t - \ell_{t-1} - b_{t-1})$, and the seasonal index of the same season $m$ time periods ago. The $s_t$ is an equation for the seasonal component. To create a predictive model based on the Holt-Winters Exponential Smoothing method, the function *Exponential Smoothing* of the statmodels library [40] is used. The function takes three parameters as input: the data set for training, the type of seasonal component add/mul, and the value of the parameter m from Equations (2) and (3)—the value of the seasonality period of the sample. The result of forecasting with the Holt-Winters model is shown in Figure A2 (see Appendix A).

### *3.3. ETS*

The ETS models are a family of time series models with an underlying state space model consisting of a level component, a trend component (T), a seasonal component (S), and an error term (E). Point forecasts can be obtained from the models by iterating the equations for $t = T + 1, \ldots, T + h$ and setting all $= 0$ for $t > T$. For example, for the model, ETS (M, A, N) $= (1 + \varepsilon_{T+1})$. Therefore $\hat{y}_{T+1|T} = \ell_T + b_T$. Similarly,

$$y_{T+2} = (\ell_{T+1} + b_{T+1})(1 + \varepsilon_{T+2}) = [(\ell_T + b_T)(1 + \alpha\varepsilon_{T+1}) + b_T + \beta(\ell_T + b_T)\varepsilon_{T+1}](1 + \varepsilon_{T+2}). \tag{5}$$

Therefore, $\hat{y}_{T+2|T} = \ell_T + 2b_T$, and so on (see Equation (5)). These forecasts are identical to the forecasts from Holt's linear method and also to those from model ETS (A, A, N). Thus, the point forecasts obtained from the method and from the two models that underlie the method are identical (assuming that the same parameter values are used). The ETS point forecasts constructed in this way are equal to the means of the forecast distributions, except for the models with multiplicative seasonality [42]. As with the Holt-Winters model, the ETS model can be obtained using the statmodels library. The *ETSModel* function takes as input parameters an array of data with time series counts, a flag defining seasonal component type add/mul, and seasonal period value (for the example under consideration seasonal_periods = 24). The *get_prediction* method of the model object, which receives as input parameters two timestamp values corresponding to the start and end time for the desired prediction horizon, is used to obtain the prediction. When using pandas, these values can be found as start and end values of the pandas dataframe object index, where the test selection of the time series is stored. The result for the ETS model is shown in Figure A3 (see Appendix A).

### 3.4. Facebook Prophet

Prophet is a procedure for forecasting time series data based on an additive regression model, where non-linear trends are fit with yearly, weekly, and daily seasonality, plus holiday effects. It works best with time series that have strong seasonal effects and several seasons of historical data. Prophet is robust to missing data and shifts in the trend, and it typically handles outliers well [43]. The main idea of the method is presented in [31]. The Prophet is an open-source library that is distributed in R and Python. The selection of the required model parameters is fully automated, so to build a model, it is sufficient to pass the input dataset for training. The *Prophet* function of the prophet library is used to create the model. Predicted parameter values can be found using the prediction model method, with a data frame of time stamps corresponding to the prediction horizon as input. The result of forecasting with the Prophet model is shown in Figure A4 (see Appendix A).

### 3.5. XGBoost

Extreme Gradient Boosting (XGBoost) is an optimized distributed gradient boosting library designed to be highly efficient, flexible, and portable. It implements machine learning algorithms under the Gradient Boosting framework. XGBoost provides a parallel tree boosting (also known as GBDT, GBM) that solves many data science problems in a fast and accurate way. The same code runs on the major distributed environment (Hadoop, SGE, MPI) and can solve problems beyond billions of examples [44].

The main idea of the method is presented in [32]. The model is created using the XGBRegressor function of the xgboost library. The function contains a relatively large number of parameters, and a complete list can be found in the documentation section of the library [45]. In our case, only four parameters were used:

- max_depth = 6—maximum tree depth for base learners;
- learning_rate = 0.05—boosting learning rate (xgb's "eta");
- n_estimators = 5000—number of gradients boosted trees (equivalent to the number of boosting rounds);
- gamma = 0.1—minimum loss reduction required to make a further partition on a leaf node of the tree.

It should be noted that, in contrast to the previously described implementations, the xgboost library can predict only one step ahead. Previous methods allowed setting the forecast horizon length in hours after training the model, after which the model output produced an estimate of the observed parameter. The XGBoost model allows one to obtain a forecast only for one step ahead (i.e., for 1 h). The aim is to obtain the forecast 48 h ahead. It is necessary to change the scheme of data preparation, which is fed to the model input. The algorithm for generating the data used to train and test the model is as follows:

1. To generate a forecast for 1 h ahead, the model's input is the historical sample for the previous 48 h. The model's output estimates the observed parameter 1 h ahead.
2. One element shifts the historical sample dataset to the left, and as a result, its length is reduced by 1. An element is placed in the place of the missing element at the end of the sample—the estimate of the observed parameter obtained at the previous step. A new parameter estimate is obtained at the model's output, corresponding to the 2-h forecast horizon.
3. Iterations continue until the desired forecast horizon is obtained.

An explanation of this algorithm for the task of obtaining a forecast horizon for the next 48 h is shown in Figure 3. A similar approach was used to prepare data for the Long Short-Term Memory model.

When testing the model, you can use actual data from the test dataset instead of parameter estimates (real test data), but to obtain a prediction in a real-time system, you will have to use parameter estimates to generate a new dataset (synthetic test data). Below, the results will be given for real and synthetic test data.

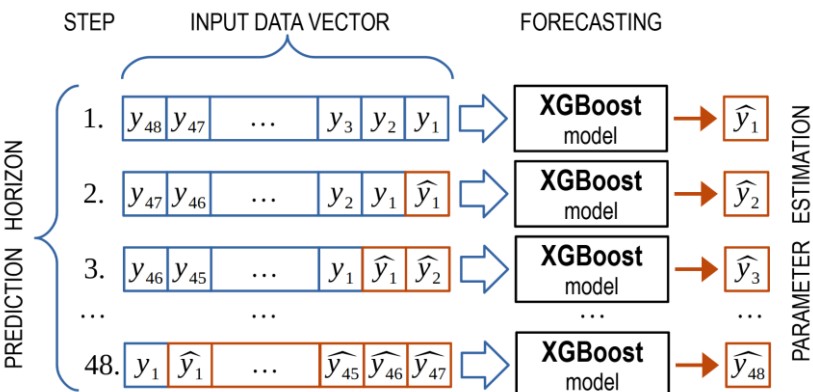

**Figure 3.** Data preprocessing algorithm for the XGBoost model.

The result of forecasting with the XGBoost model based on real and synthetic test data is shown in Figures A5 and A6 (see Appendix A).

As follows from the graphical results, we can conclude that, when using synthetic test data (i.e., the system works in real conditions), the accuracy will decrease as the forecast horizon increases. That is why we should either increase the amount of data in the input vector fed into the model or decrease the forecast horizon. The accuracy of the model will also be affected by the XGBoost model setup.

### 3.6. Long Short-Term Memory

Long Short-Term Memory (LSTM) is an artificial neural network in artificial intelligence and deep learning. Unlike standard feedforward neural networks, LSTM has feedback connections. Such a recurrent neural network (RNN) can process single data points and entire data sequences. This characteristic makes LSTM networks ideal for processing and predicting data, including time series. The main idea of the method is presented in [21].

To build the LSTM model, we used the keras library and a set of functions *Sequential*, *Dense*, and LSTM, which were used to set the structure of the neural network. The following parameters were chosen to build the model:

- batch_size = 16—number of samples per gradient update;
- epochs = 200—number of epochs to train the model;
- units = 32—dimensionality of the output space.

A description of the LTSM model obtained using the keras library is shown below.

```
LSTM model summary
Model: "sequential"
```

| Layer (type) | Output Shape | Param # |
|---|---|---|
| lstm (LSTM) | (None, 32) | 4352 |
| dense (Dense) | (None, 1) | 33 |

```
Total params: 4,385
Trainable params: 4,385
Non-trainable params: 0
```

The result of forecasting with the LSTM model based on real and synthetic test data is shown in Figures A7 and A8 (see Appendix A).

### 4. Results and Discussion

Table 2 shows the results of calculating statistical metrics to assess the accuracy of the created models. The first columns are the statistical metrics, T.Time is the time in seconds that it took to build the model, and P.Time is the time in seconds to calculate the prediction.

The build model time includes the time to obtain all necessary data, optimal parameter selection, and model time creation with the Python libs.

**Table 2.** Statistical and performance time analysis.

| Model | $R^2$ | MSE | RMSE | MAE | MAPE | T.Time | P.Time |
|---|---|---|---|---|---|---|---|
| SARIMA | 0.961 | 0.076 | 0.276 | 0.198 | 0.035 | 1298.060 | 0.020 |
| Holt-Winters ES | 0.921 | 0.156 | 0.396 | 0.324 | 0.059 | 0.049 | 0.001 |
| ETS | 0.945 | 0.109 | 0.329 | 0.254 | 0.043 | 0.285 | 0.001 |
| Prophet | 0.918 | 0.162 | 0.402 | 0.331 | 0.062 | 0.881 | 0.754 |
| XGBoost (real test data) | 0.975 | 0.050 | 0.224 | 0.163 | 0.029 | 7.505 | 0.005 |
| XGBoost (synthetic test data) | 0.954 | 0.091 | 0.301 | 0.228 | 0.043 | 7.505 | 0.235 |
| LSTM (real test data) | 0.960 | 0.080 | 0.282 | 0.218 | 0.041 | 39.505 | 0.361 |
| LSTM (synthetic test data) | 0.907 | 0.184 | 0.429 | 0.322 | 0.063 | 39.505 | 3.185 |

All models show good quality with a good R-squared metric > 0.9. The models with R-squared > 0.95 shows excellent accuracy. However, the second factor—time for preprocessing and training models and time to obtain the forecasting—will be used to choose the best model.

The leaders in accuracy are three models: SARIMA, LSTM, and XGBoost. However, SARIMA and LSTM significantly lose the time required to create a new model, which is caused by the automatic procedure of selecting parameters. The SARIMA method has high accuracy, but its application in systems with non-stationary time series with seasonal variability can be complex because of the long time required for automatic model preparation. The LSTM method showed promising results on accurate data. Still, the accuracy decreases significantly on synthetic test data, i.e., the forecast horizon for natural systems will decrease to preserve the precision. This indicates that the method has good potential but requires more careful tuning, network structure choice, and input data optimization for training and model use. XGBoost requires less than 10 s for fully automatic model building. The time it takes to obtain a prediction when the control signal is applied once per second is 0.2 s, i.e., about 20%. In addition to Python, the XGBoost libraries are implemented in the C++/C# programming languages, making it easy to bring it into production. Therefore, the optimal choice for solving the task is the XGBoost method.

It should be noted that this study has limitations: we assume that the sensors are working correctly. New data are always coming in. If the system is interrupted for more than two days, it will be necessary to accumulate new continuous historical data for the last two to three weeks before starting.

## 5. Conclusions

The article considers various methods of forecasting time series with seasonality. Classical and newer approaches based on machine learning algorithms and neural networks, as well as statistical models, are presented. A test of prediction accuracy using statistical metrics was performed for each of the considered methods. The authors proposed an approach that allows using small historical data samples (less than 21 days) to build a short-term forecast (nearest 48 h). It reduces the load on computing resources and maintains the accuracy of the estimates. It is crucial when implementing predictive systems based on embedded devices that implement real-time automatic control. The model must be retrained once a day to achieve the goal, considering the newly received data. Thus, the criterion responsible for the time for collecting data, training the model, and obtaining a forecast plays a decisive role in choosing a model. The best method, according to a set of attributes, turned out to be the XGBoost method. Nevertheless, optimizing algorithms to search model parameters can bring other algorithms, such as LSTM, to the forefront. Further research will be devoted to the optimization of model learning. A new approach for the hyperparameters finding procedure and a new algorithm for the optimal data length calculation should be developed. The existing methods will be extended for the using of

the recommended systems and robust optimal control real-time systems. The participation of the operator in setting up the algorithms should be minimized.

**Author Contributions:** Conceptualization, V.K. and V.A.; methodology, V.K.; software, V.A.; formal analysis, V.K.; investigation, V.K. and V.A.; writing—original draft preparation, V.A.; writing—review and editing, V.K.; visualization, V.A.; supervision, V.K.; All authors have read and agreed to the published version of the manuscript.

**Funding:** This research received no external funding.

**Institutional Review Board Statement:** Not applicable.

**Informed Consent Statement:** Not applicable.

**Data Availability Statement:** The code and data presented in this study available on GitHub "https:// github.com/vasoftlab/algorithms-2263871 (accessed on 8 May 2023)". All details can be provided upon request from the corresponding author.

**Conflicts of Interest:** The authors declare no conflict of interest.

## Abbreviations

The following abbreviations are used in this manuscript:

| | |
|---|---|
| ADfuller | Augmented Dickey-Fuller |
| ANN | Artificial Neural Network |
| ARIMA | Autoregressive integrated moving average |
| DB | Data base |
| ETS | Exponential smoothing |
| LSTM | Long short-term memory |
| MAE | Mean absolute error |
| MAPE | Mean absolute percentage error |
| MSE | Mean squared error |
| OPC | Open Platform Communications |
| P.Time | Prediction time |
| R2 | Correlation coefficient |
| RMSE | Root mean squared error |
| RNN | Recurrent neural network |
| SARIMA | Seasonal autoregressive integrated moving average |
| T.Time | Training time |

## Appendix A

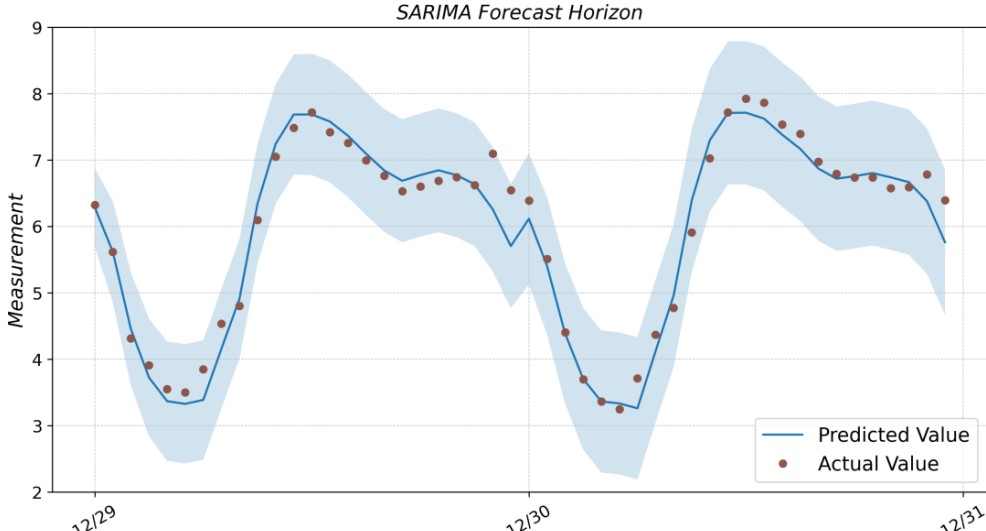

**Figure A1.** Prediction result with SARIMA model and actual values from the test dataset.

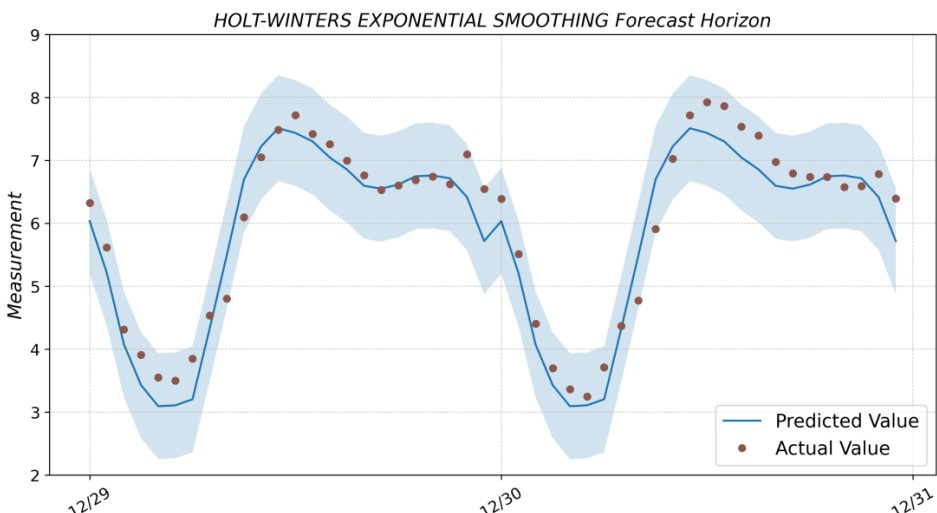

**Figure A2.** Prediction result with Holt-Winters Exponential Smoothing model and actual values from the test dataset.

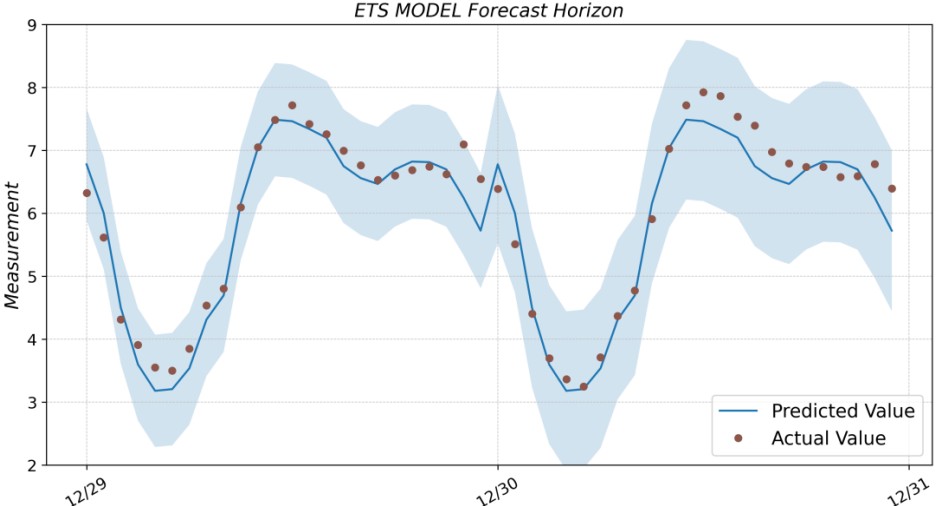

**Figure A3.** Prediction result with ETS model and actual values from the test dataset.

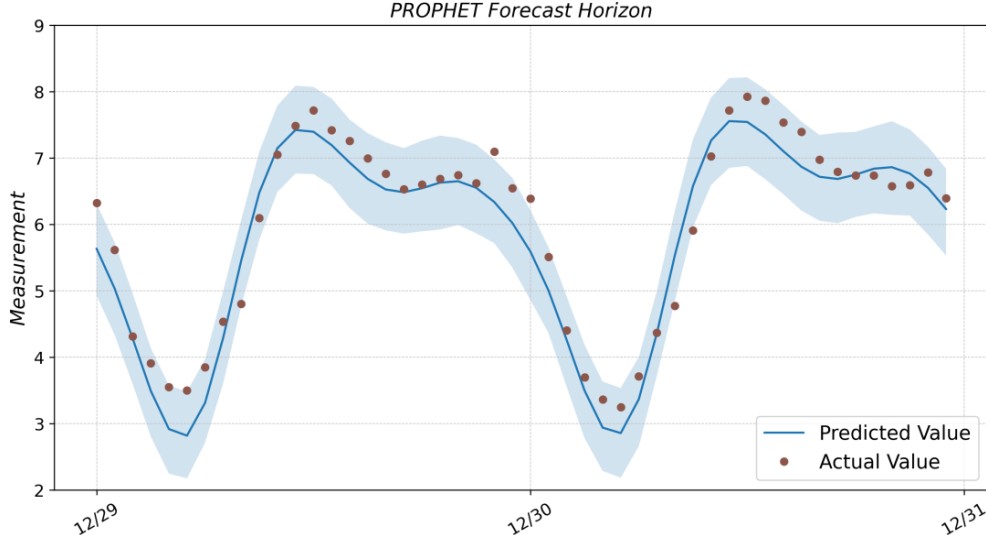

**Figure A4.** Prediction result with Prophet model and actual values from the test dataset.

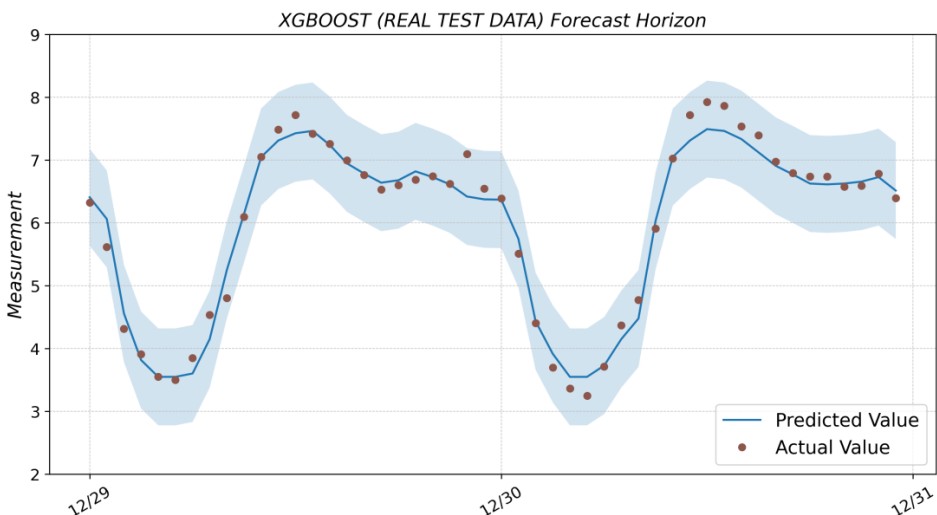

**Figure A5.** Prediction result with the XGBoost model (based on real test data) and actual values from the test dataset.

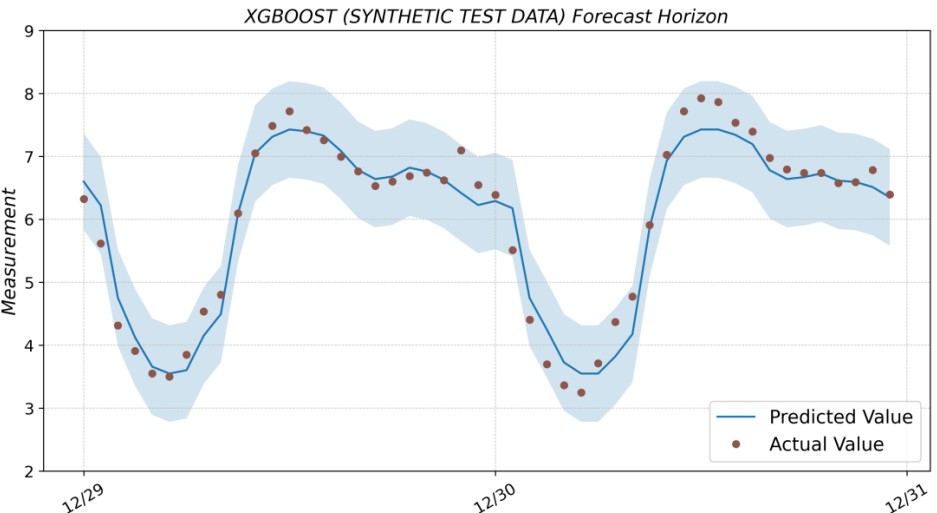

**Figure A6.** Prediction result with the XGBoost model (based on synthetic test data) and actual values from the test dataset.

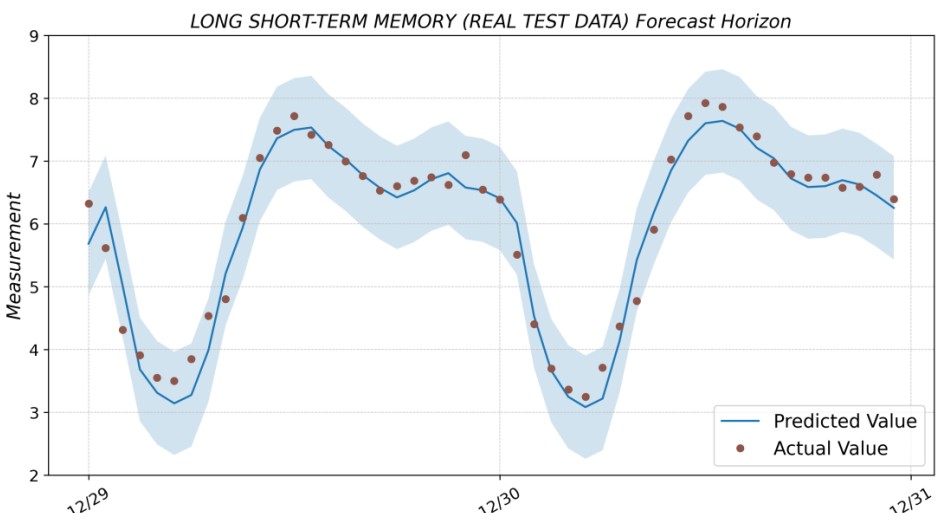

**Figure A7.** Prediction result with the LSTM model (based on real test data) and actual values from the test dataset.

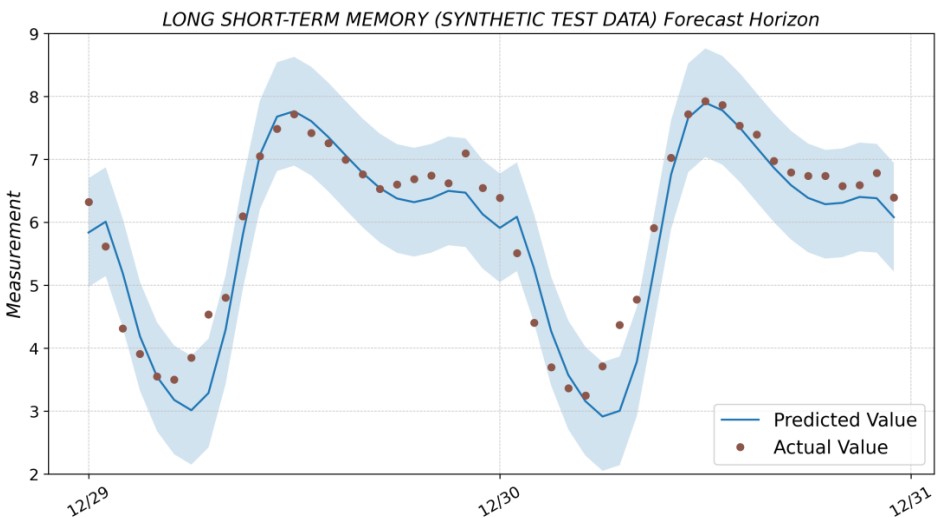

**Figure A8.** Prediction result with the LSTM model (based on synthetic test data) and actual values from the test dataset.

## Appendix B

| **Algorithm A1:** XGBoost model retraining and prediction (48 h ahead) |
|---|
| 1    $time_{start}$; $time_{stop}$ ← *The time interval for the training data* |
| 2    $data\_preprocessing(time_{start}, time_{stop})$ ← *Resample and training data generation* |
| 3    $Output:$ $X\_train$, $y\_train$ |
| 4    $max_{depth} = 6$ ← *Maximum tree depth for base learners* |
| 5    $learning_{rate} = 0.05$ ← *Boosting learning rate* |
| 6    $n_{estimators} = 5000$ ← *Number of gradients boosted trees* |
| 7    $gamma = 0.1$ ← *Minimum loss reduction* |
| 8    $model = XGBRegressor(objective =' reg:squarederror',$ ← *XGBoost regressor* |
| 9    $n\_estimators = n\_estimators,$ |
| 10   $max\_depth = max\_depth,$ |
| 11   $learning\_rate = learning\_rate,$ |
| 12   $gamma = gamma)$ |
| 13   $model.fit(X\_train, y\_train)$ ← *Training model* |
| 14   $y\_pred\_new = [\,]$ ← *Prediction array initialization* |
| 15   for $i = (1\ to\ 48)$ do |
| 9    $x\_list = x\_train.iloc[:, 1:].values.tolist()$ |
| 10   $y\_list = y\_pred.tolist()$ |
| 11   $x\_train = pd.DataFrame(x\_list[0]+y\_list[0])$ ← *New train data* |
| 12   $x\_train = x\_train.transpose()$ |
| 13   $y\_pred = model.predict(x\_train)$ ← *Get prediction* |
| 14   $y\_pred = pd.DataFrame(y\_pred)$ ← *Convert to the Pandas dataframe* |
| 15   $y\_pred = y\_pred.values$ |
| 16   $y\_pred\_new.append(y\_pred.tolist()[0])$ ← *Add new predicted value* |
| 17   end for |
| 18   $Output:$ $y\_pred\_new$ ← *Prediction dataset for the next 48 hrs* |

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
