# Peer review of "Time-Series Forecasting of Seasonal Data Using Machine Learning Methods"

_algorithms, doi:10.3390/a16050248_

Round 1

Reviewer 1 Report

This study considers constructing a time series forecasting model with seasonal variability. Several methods including SARIMA, Holt-Winters Exponential Smoothing, ETS, Facebook Prophet, XGBoost, and Long Short-Term Memory are used. The paper is written very poorly. However, before deciding on the paper, I would like to give a chance to the authors to improve the paper. I will give my decision on the revised version of the paper. My specific comments are as under: 

  1. The abstract is written very poorly. It does not convey what the paper is about and what is the novelty of this work. The complete abstract must be rewritten in no more than 150 words.
  2. In the abstract, nothing has been mentioned which data is used to compare different models. No motivation has been given why to compare different models for some specific task.
  3. At the end of the abstract, please give the numerical results of the study in no more than two lines.
  4. Line 32, in the equation, (t) should be only t in the subscript of the variables.
  5. Avoid lump-sum references in the introduction section.
  6. The author must provide the details of the cited works. What they have done and what were the shortcomings of their study? How this study will address those shortcomings?
  7. The literature should be updated and expended. Especially those works must be considered that account for different seasonality components. For example, functional data approach for short-term electricity demand forecasting, Modeling and forecasting electricity demand and prices: a comparison of alternative approaches, etc.
  8. Please provide a section-wise breakup at the end of section 1.
  9. Highlight the novelty of current work in the introduction section, point-by-point.
  10. L110-121. The data should be placed in a proper table with a proper caption.
  11. The authors should provide a link for the retrieval of data.
  12. The training data period is too short. It must be more than six months at least.
  13. The testing period is too short. The author should use at least a month testing period and compute the average error summaries for the whole testing set.
  14. The output on page 5 shows that the authors have no experience in academic writing. Convert all the results into proper Tables. Give them proper captions. Remove all unnecessary details from the tables.
  15. Very poor writing from page 5 onwards. The authors should start a methodology section first and define all the models in this section with their estimation procedure.
  16. The next section should be analysis and results where the authors should discuss the results of each model. 
  17. Currently, the paper is not readable and hence the paper first must be restructured. Once it has been done, I will provide further comments on the paper. 
  18. Please add the study limitations and future recommendations to the conclusion section.

Author Response

Response to Reviewer 1 Comments

The authors are grateful for the valuable comments that made it possible to improve the paper.

Here are the responses to the comments.

Point 1: The abstract is written very poorly. It does not convey what the paper is about and what is the novelty of this work. The complete abstract must be rewritten in no more than 150 words.

Response 1: Corrected. The abstract has been rewritten, and the number of words has been reduced. A brief description of the novelty of the work and main results has been added. Lines 8-18

Point 2: In the abstract, nothing has been mentioned which data is used to compare different models. No motivation has been given why to compare different models for some specific task.

Response 2: Corrected. The abstract has been rewritten. The description of the task's specific added. The data used for the models described. Lines 8-18

Point 3: At the end of the abstract, please give the numerical results of the study in no more than two lines.

Response 3: Numerical results added. Lines 8-18

Point 4: Line 32, in the equation, (t) should be only t in the subscript of the variables.

Response 4: Corrected. Lines 30-31 

Point 5: Avoid lump-sum references in the introduction section.

Response 5: The links in the introductory part were used only to show in which areas seasonal time series forecasting problems are applied. The methods and algorithms of all approaches are generally the same, so the works were mentioned once. Thanks for the note.

Point 6: The author must provide the details of the cited works. What they have done and what were the shortcomings of their study? How this study will address those shortcomings?

Response 6: See the previous answer. We did not use the results from most of the cited works. Based on limited historical data, we used the same methods in a specific manner to get the accuracy for the prediction models of influent for the waste treatment plant. Cited articles used the time-series predictions in fully different field.

Point 7: The literature should be updated and expended. Especially those works must be considered that account for different seasonality components. For example, functional data approach for short-term electricity demand forecasting, Modeling and forecasting electricity demand and prices: a comparison of alternative approaches, etc.

Response 7: Thanks for the note.

Point 8: Please provide a section-wise breakup at the end of section 1.

Response 8: Corrected. Added to the end of the Introduction section. Lines 88-98

Point 9: Highlight the novelty of current work in the introduction section, point-by-point.

Response 9: Corrected. Added to the end of the Introduction section. Lines 77-87

Point 10: L110-121. The data should be placed in a proper table with a proper caption.

Response 10: Corrected. Lines 129-130

 Point 11: The authors should provide a link for the retrieval of data.

Response 11: Corrected. The link to the GitHub repository added (see Data Availability Statement). The folder data contains all data used in this research. Line 453

Point 12: The training data period is too short. It must be more than six months at least.

Response 12: No, it’s not necessary. Moreover, it was a request from our technical partner to get an algorithm that can use a short dataset for the model training. Preservation of accuracy is achieved through constant retraining of the model (once a day). The same for the testing period – the forecasting horizon should be 48 hours only due to the specification. Our algorithm implementation has been working in a real-time control system for over a year, saving the client more than twice the electricity cost. So, the quality of the algorithm is confirmed with the long period real-time experiment.

Point 13: The testing period is too short. The author should use at least a month testing period and compute the average error summaries for the whole testing set.

Response 13: See the previous answer.

Point 14: The output on page 5 shows that the authors have no experience in academic writing. Convert all the results into proper Tables. Give them proper captions. Remove all unnecessary details from the tables.

Response 14: The output on page 5 is not a table. Instead, it is logged data from the training procedure. It shows that the method SARIMA takes a lot of time to train the model. Also, we added log output for ADfuller test that confirmed the seasonality of the dataset. All this information can be used as information for researchers that will try to reproduce the experiment and will need comments on all outputs. It's a part of our experiment, so we don't want to remove it.

Point 15: Very poor writing from page 5 onwards. The authors should start a methodology section first and define all the models in this section with their estimation procedure.

Response 15: Corrected. The “Methodology” section used to describe the general steps to be done to get the prediction model. All models used in research are in section “Development of the models”. The estimation procedure (see below) is a standard statistical metrics.

Point 16: The next section should be analysis and results where the authors should discuss the results of each model.

Response 16: All results are discussed in section Results and Discussions. The objective function for each model is accuracy and time for training and forecasting. These results are in Table 2. Also, we choose the best model by the combination of these factors. See lines 409-425

Point 17: Currently, the paper is not readable and hence the paper first must be restructured. Once it has been done, I will provide further comments on the paper.

Response 17: Thanks for the note.

Point 18: Please add the study limitations and future recommendations to the conclusion section.

Response 18: Corrected. Added. Lines 426-429

Reviewer 2 Report

Dear Authors!

The work is meaningful, and is of interest to specialists in the fields of modeling and development of control systems.

Corresponds to the theme of the special issue and magazine.

I have no questions about the content as a whole, tk. I am an expert in this field, and the course of your thoughts is clear to me.

However, there are several suggestions that could help improve the quality of the work and generate more reader interest:

1. There should be at least 5 keywords. This will significantly expand the distribution of the manuscript during its indexing.

2. I highly recommend giving the presentation of the algorithm presented in the classical way (in any traditional way). I assure you it would be useful for the audience of the magazine.

3. The Results Section should with methodological results.

4. The Discussion Section should estimate approaches, described in the Introduction Section.

5. The quality of some graphic materials leaves much to be desired. I'm sure it's up to you.

Good luck with revisions!

Author Response

Response to Reviewer 2 Comments

The authors are grateful for the valuable comments that made it possible to improve the paper.

Here are the responses to the comments.

Point 1: There should be at least 5 keywords. This will significantly expand the distribution of the manuscript during its indexing.

Response 1: Corrected. Lines 19-20

Point 2: I highly recommend giving the presentation of the algorithm presented in the classical way (in any traditional way). I assure you it would be useful for the audience of the magazine.

Response 2: Corrected. The algorithm added to Appendix B. Line 480

Also the link to the GitHub repository with the full source codes and test data added. Line 453

Point 3: The Results Section should with methodological results.

Response 3: Corrected. Main results (including methodological) are in lines 409-425

Point 4: The Discussion Section should estimate approaches, described in the Introduction Section.

Response 4: Corrected. The methods estimation is in lines 409-425

Point 5: The quality of some graphic materials leaves much to be desired. I'm sure it's up to you.

Response 5: Corrected. All images used for the article have a DPI = 300 (we used the matplotlib set_dpi(300) function in our charts generation script). All other images were exported to the graphical editor and also saved with DPI = 300. But it’s possible the web-version of MDPI system compress images while exporting to the PDF. Unfortunately, we can’t affect on this.

Reviewer 3 Report

Dear Authors, Please carefully check and solve all observations formulated (below) and sent to the editors of the Algorithm MDPI journal to which you submitted your manuscript:

<<Dear Editors,

As the authors state, this paper (id algorithms-2263871, entitled “Time-series forecasting of seasonal data using machine learning methods”) considers constructing a time series forecasting model with seasonal variability, describing a technological process. The authors claim that the general methodology of model building is described, as well as the requirements for the input data sets. They also state that several methods are compared: SARIMA, Holt-Winters Exponential Smoothing, ETS, Facebook Prophet, XGBoost, and Long Short-Term Memory. The authors additionally claim that a comparative analysis of the accuracy of models is made based on obtained statistical metrics and the performance of the methods was also analyzed. The best method for solving the task, as they state,  has been chosen according to the sum of estimations. In conclusion, recommendations for improving prediction accuracy are given, and directions for future research in this area are outlined.

After reading this paper, I think there are still some issues waiting to be dealt with.

I will start with the format ones. Then I will continue with those related to this paper’s substance if applicable.

I mention below the following:

  • Turnitin indicates a similarity score of 27 (if removing references) and 33 (with references included) which I think is acceptable;

  • English language and style issues - Grammarly (https://app.grammarly.com) on default settings (American English, Set Goals: Audience=Knowledgeable, Formality=Neutral, Domain=General) detected only for the text block resulting from the concatenation of Title+Abstract+Keywords +Conclusion(s): Only 4 correctness issues / critical alerts
    but many (24) complex ones / advanced suggestions.

Consequently, the resulting Grammarly overall/total score as reported by this online tool was just 76 (right edge of Fair i.e. >=75, but not Good i.e >=80, or Very Good/Excellent i.e. >=90) out of 100 (max) for this four-component sample above. Moreover, since the authors do not appear to be native English speakers, I suggest a comprehensive revision of the English language and style for the entire article using Grammarly or another specialized tool (full English language and style report for the entire paper required);

  • The paper must follow all the instructions of the journal precisely indicated at: https://www.mdpi.com/journal/algorithms/instructions;

  • All references to equations/formulas must be explicitly and precisely formulated in the main text (e.g., see eq. x);

  • The authors must avoid ending some sections/subsections with figures, tables, formulas, or other components (e.g., eq.Fig.4 just before subsection 3.3, and no explanatory text after). The authors are required to check the entire manuscript for similar issues;

  • The authors must additionally ensure that all figures have the required resolution (minimum 1000 pixels width/height, or a resolution of 300 dpi or higher). Most of the figures suffer in these terms. The .pdf export options (image compression ones) must be verified too;

  • The legend texts/titles of some of the figures are too large (more than one line). The authors should move part of these text blocks in the main text of the manuscript near the first reference to them. The authors are also required to check the entire manuscript for such issues;

  • There is a high number of figures (11) in this manuscript. Some of them which are considered by the authors not essential for understanding the main flow of ideas in the manuscript must be moved to the Appendix section. If this section is not existing, the authors must create one; 

  • There is no room for equations in Results and Discussions. They must be moved to the Data and Methods/Methods section;

  • The List of Abbreviations at the end of the manuscript is needed;

  • The same for the Limitations corresponding to the research approach used;

  • The authors are also required to include more explanations and precise details about the standard view of the accuracy values (>=70% and <80%-fair models; >=80% and <90%-good models; >=90%-very good/excellent models) and their application here. They should provide more references to scientific papers where this topic is considered and the accuracy intervals are precisely defined;

  • They should additionally indicate (supporting references to journal papers also needed) other accuracy measures for time series except the ones indicated and used in their approach;

  • The final list of 31 references (from which, I counted 4 conference papers that are not considered finalized research work) is not enough. It indicates as very probable that many important related contributions in journal papers have not been cited in the Related Work/Literature Review section;

  • Moreover, the section dedicated to the interpretation of the results (Results and Discussions - not clearly emphasized this way) needs more development and cited references to similar/different results (already published articles in highly rated scientific journals);

  • The authors must understand that replicability as a fundamental principle in science (https://doi.org/10.1007/s10516-021-09610-2) starts with data and it is not a fad but a necessity. Therefore, they should insert in the Data Availability Statement section at the end of the manuscript all precise links to all data providers’ / own datasets supporting this manuscript;

  • Following this scientific principle above, the authors should provide full details about the software (including the precise name of the provider and version number of all the tools/apps) and also complete details about the hardware they used to test their approach and obtain the results presented in this manuscript;

  • Following the same replicability principle (https://doi.org/10.1038/nature.2016.20504), if some own algorithms have been used, the authors must precisely identify them in their own GitHub (or similar) repository section. If not existing, the authors must create one for the entire project corresponding to this manuscript. Otherwise, the authors must clearly specify that no custom algorithms have been used;

  • The authors must also triangulate using many methods and techniques (https://doi.org/10.1038/d41586-018-01023-3) and perform many rounds of random cross-validations (https://doi.org/10.1007/978-0-387-39940-9_565, https://doi.org/10.1016/j.procs.2021.08.128) in order to prove their approach and the corresponding research results obtained are robust;

  • The Conclusions section should be restructured in a way that would better emphasize the authors’ contributions.

Thank you for the opportunity to read and check this manuscript!>>

Author Response

Response to Reviewer 3 Comments

The authors are grateful for the valuable comments that made it possible to improve the paper.

Here are the responses to the comments.

Point 1: Turnitin indicates a similarity score of 27 (if removing references) and 33 (with references included) which I think is acceptable;

English language and style issues - Grammarly (https://app.grammarly.com) on default settings (American English, Set Goals: Audience=Knowledgeable, Formality=Neutral, Domain=General) detected only for the text block resulting from the concatenation of Title+Abstract+Keywords +Conclusion(s): Only 4 correctness issues / critical alerts
but many (24) complex ones / advanced suggestions.

Consequently, the resulting Grammarly overall/total score as reported by this online tool was just 76 (right edge of Fair i.e. >=75, but not Good i.e >=80, or Very Good/Excellent i.e. >=90) out of 100 (max) for this four-component sample above. Moreover, since the authors do not appear to be native English speakers, I suggest a comprehensive revision of the English language and style for the entire article using Grammarly or another specialized tool (full English language and style report for the entire paper required);

The paper must follow all the instructions of the journal precisely indicated at: https://www.mdpi.com/journal/algorithms/instructions;

Response 1: Corrected. The article structure fixed. The score of the Grammarly (the text without images, formals, tables) is 94. We are using Grammarly with the Premium account. Language preference – American English (see attach file).

Point 2: All references to equations/formulas must be explicitly and precisely formulated in the main text (e.g., see eq. x);

Response 2: Corrected

Point 3: The authors must avoid ending some sections/subsections with figures, tables, formulas, or other components (e.g., eq.Fig.4 just before subsection 3.3, and no explanatory text after). The authors are required to check the entire manuscript for similar issues;

Response 3: Corrected. All images moved to the Appendix A.

Point 4: The authors must additionally ensure that all figures have the required resolution (minimum 1000 pixels width/height, or a resolution of 300 dpi or higher). Most of the figures suffer in these terms. The .pdf export options (image compression ones) must be verified too

Response 4: All figures have dpi = 300. Probably while the automatic converting with the online MDPI service for the editors, the images were compressed. But we can’t influence it. All our drawings in manuscript in doc format comply with the requirements of the publisher

Point 5: The legend texts/titles of some of the figures are too large (more than one line). The authors should move part of these text blocks in the main text of the manuscript near the first reference to them. The authors are also required to check the entire manuscript for such issues;

Response 5: All figures’ titles are one-line text. Some legends are two lines of text. But if we use one-line text legends, it will hide the data lines, for us more important to show the data points. So, the figure's secondary elements have not changed. But we will take this remark into account for the preparation of other illustrative materials.

Point 6: There is a high number of figures (11) in this manuscript. Some of them which are considered by the authors not essential for understanding the main flow of ideas in the manuscript must be moved to the Appendix section. If this section is not existing, the authors must create one;

Response 6: Corrected. All figures with the method validation results moved to the Appendix A. Line 453-475

Point 7: There is no room for equations in Results and Discussions. They must be moved to the Data and Methods/Methods section;

Response 7: Corrected. The equation (5) → (1) moved to the Methodology section. Line 169

Point 8: The List of Abbreviations at the end of the manuscript is needed;

Response 8: Corrected. Added List of Abbreviations in Appendix C. Lines 483-

Point 9: The same for the Limitations corresponding to the research approach used;

Response 9: Corrected. Lines 426-429

Point 10: The authors are also required to include more explanations and precise details about the standard view of the accuracy values (>=70% and <80%-fair models; >=80% and <90%-good models; >=90%-very good/excellent models) and their application here. They should provide more references to scientific papers where this topic is considered and the accuracy intervals are precisely defined;

Response 10: Added in Results and Discussions section. Lines 409-425

 Point 11: They should additionally indicate (supporting references to journal papers also needed) other accuracy measures for time series except the ones indicated and used in their approach;

Response 11: We don’t need other metrics. The main metrics we used to check the model accuracy is r-squared. Furthermore, all other citied works used exactly the same set of statistical metrics for the accuracy calculation.

Point 12: The final list of 31 references (from which, I counted 4 conference papers that are not considered finalized research work) is not enough. It indicates as very probable that many important related contributions in journal papers have not been cited in the Related Work/Literature Review section;

Response 12: Most of citied works (including the conference papers) are used for only to show where the time-series forecasting is used. We used own algorithm that based on methods that have been recognized for a long time, but were not involved in the problem we solved. All links for these methods are in the reference list. Only articles published in public access were used in this article (see the ResearchGate).

Point 13: Moreover, the section dedicated to the interpretation of the results (Results and Discussions - not clearly emphasized this way) needs more development and cited references to similar/different results (already published articles in highly rated scientific journals);

Response 13: See the previous answer.

Point 14: The authors must understand that replicability as a fundamental principle in science (https://doi.org/10.1007/s10516-021-09610-2) starts with data and it is not a fad but a necessity. Therefore, they should insert in the Data Availability Statement section at the end of the manuscript all precise links to all data providers’ / own datasets supporting this manuscript;

Response 14: The data were provided by the administration of the treatment plant, where our algorithm has been working for more than one year. The full database has not public access. But we created the repository with the python code and added the data folder with the data table that was used for this article. The GitHub link has been added to the Data Availability Statement. So, all results presented in the article can be reproduced. Line 453

Point 15: Following this scientific principle above, the authors should provide full details about the software (including the precise name of the provider and version number of all the tools/apps) and also complete details about the hardware they used to test their approach and obtain the results presented in this manuscript;

Response 15: Corrected. The information about software and libraries versions added to the Results and Discussions section. Lines 387-394

Point 16: Following the same replicability principle (https://doi.org/10.1038/nature.2016.20504), if some own algorithms have been used, the authors must precisely identify them in their own GitHub (or similar) repository section. If not existing, the authors must create one for the entire project corresponding to this manuscript. Otherwise, the authors must clearly specify that no custom algorithms have been used;

Response 16: The link to the GitHub repository added to the Data Availability Statement section. Line 453

Point 17: The authors must also triangulate using many methods and techniques (https://doi.org/10.1038/d41586-018-01023-3) and perform many rounds of random cross-validations (https://doi.org/10.1007/978-0-387-39940-9_565, https://doi.org/10.1016/j.procs.2021.08.128) in order to prove their approach and the corresponding research results obtained are robust;

Response 17: Thanks for the note. But the cross-validation can’t be used for the ordered time series. As we mention below, the described approach and algorithm implemented in integrated in build-in real-time control system on real treatment plant. The system works more then one year and our client already saved the electricity resources twice. So, the robust confirmed be experimental usage and the main aim for this task is a model accuracy and the time that used for training and forecasting.

Reference https://scikit-learn.org/stable/modules/cross_validation.html

The cross-validation will change the data order, that’s not possible for time-series which ordered by the timestamp. The nature of seasonality, in this case, will be violated. The cross-validation on a rolling basis also not applied for the data with seasonality. At least for our case when we trying to get short term forecast (see attach file).

Point 18: The Conclusions section should be restructured in a way that would better emphasize the authors’ contributions.

Response 18: Corrected. Added the authors’ contributions Lines 427-440

Reviewer 4 Report

In this research machine learning methods are applied for the analysis of seasonal time series. The forecast results are used in real time control systems.

The innovative aspects of this research are not highlighted. Authors state that "the novelty of the work lies in the use of machine learning methods for predicting the technological process", but this is not an innovative aspect as a lot of machine learning seasonal time series methods have been proposed in the literature. Therefore, it seems that this study is more a survey than a new machine learning approach applied to seasonal time series.

ARIMA and Holt-Winters Exponential Smoothing are well known time series forecasting statistical models, they are not machine learning forecasting models.

The structure of the sections should be changed. The discussion on time series forecasting models should be moved from section 3 to a previous section (e.g. 2. Preliminary), moving the Methodology section to a later section (e.g. 3. Methodology).

Figure 2 should be inserted at the beginning of the Methodology section, so that the authors can briefly describe the adopted methodology schematized in Fig. 2 and only then describe the individual processes.

The results in Tab. 2 show that the model building time (T.Time) required for SARIMA is significantly higher than that required for building other models (1298 sec). How do the authors explain this result? What does it come from?

In the conclusions, the authors must describe in more detail what the future prospects of the research are, rather than summarily stating "Further research will be devoted to the optimization of model learning".

There are several grammatical errors in the text that need to be corrected.

Author Response

The authors are grateful for the valuable comments that made it possible to improve the paper.

Here are the responses to the comments.

Point 1: The innovative aspects of this research are not highlighted. Authors state that "the novelty of the work lies in the use of machine learning methods for predicting the technological process", but this is not an innovative aspect as a lot of machine learning seasonal time series methods have been proposed in the literature. Therefore, it seems that this study is more a survey than a new machine learning approach applied to seasonal time series.

Response 1: We added some more clarifications at the beginning of the Methodology section. The task of our study was not to develop a new algorithm but to find an optimal approach that can be used for existing algorithms. This article describes the solution of a real practical problem and not a review of existing methods, which may be helpful to engineers and researchers in this field.

As a result, computing resources are minimized, the system is based on the built equipment, the justification for predicting a catastrophe, the algorithm for using robust properties, and not requiring adjustments when using external factors. Lines 99-110

Point 2: ARIMA and Holt-Winters Exponential Smoothing are well known time series forecasting statistical models, they are not machine learning forecasting models.

Response 2: We agree. That’s why we classified it as “classical approach” (see Introduction section.) This is our mistake in wording. Nevertheless, these methods give qualitative predictions for our problem. The only downside is the time cost. Lines 62-67

Point 3: The structure of the sections should be changed. The discussion on time series forecasting models should be moved from section 3 to a previous section (e.g. 2. Preliminary), moving the Methodology section to a later section (e.g. 3. Methodology).

Response 3: Thank you for the note; our original structure was as you described, and we changed it due to the recommendations of other reviewers. That's why it exists in the current format.

Point 4: Figure 2 should be inserted at the beginning of the Methodology section, so that the authors can briefly describe the adopted methodology schematized in Fig. 2 and only then describe the individual processes.

Response 4: Fixed. The Fig 2. (new name is Fig. 1.) moved to the beginning of the Methodology section. Lines 120-148

Point 5: The results in Tab. 2 show that the model building time (T.Time) required for SARIMA is significantly higher than that required for building other models (1298 sec). How do the authors explain this result? What does it come from?

Response 5: As we discussed above, one of the main factors for the algorithm is the time factor. The predictive model is constantly retrained. The time factor consists of two parts - the time for preparing data for models and the time for obtaining a forecast.

Standard Python functions are used for implementation. Implementation details can be found in the code repository

 https://github.com/vasoftlab/algorithms-2263871

For example, to calculate the time to receive a forecast using SARIMA, use the code

m = SARIMAX(data, order=(3, 0, 2), seasonal_order=(0, 1, 2, 24),

            enforce_stationarity=False, enforce_invertibility=False).fit()

start_time = time.time()

forecast = m.get_forecast(steps=forecast_horizon, signal_only=True)

pred_time = f'{(time.time() - start_time): .3f}

We used a workstation running the Windows 10 operating system for all calculations. Workstation hardware: Intel Core i9-9900K CPU, 32 GB RAM, Samsung 970 PRO 512Gb SSD.

Of course, the time may differ on a different machine, but the resulting table allows us to conclude which of the algorithms calculates the forecast faster.

Point 6: In the conclusions, the authors must describe in more detail what the future prospects of the research are, rather than summarily stating "Further research will be devoted to the optimization of model learning".

Response 6: The Conclusions section corrected. We added list of tasks that should be solved in further research (to the end of the Conclusions section). Lines 428-434

Point 7: There are several grammatical errors in the text that need to be corrected.

Response 7: Corrected. The score of the Grammarly (the text without images, formals, tables) is 94. We are using Grammarly with the Premium account. Language preference – American English.

Reviewer 5 Report

This manuscript investigated machine learning methods for time-series prediction of seasonal data, which is a revised version. I can see that this manuscript has been significantly modified based on the reviewers’ comments. Accordingly, I suggest that it can be accepted for publication in Algorithms, after a minor revision by addressing the following comments.

1.       The main innovation and contribution of this research should be clearly clarified in abstract and introduction.

2.       Broaden and update literature review on machine learning in data analysis application. E.g. Torsional capacity evaluation of RC beams using an improved bird swarm algorithm optimised 2D convolutional neural network; Nonlinear hysteretic parameter identification using an attention-based long short-term memory network and principal component analysis.

3.       Please add the information about how the authors select the optimal hyperparameters of machine learning models to achieve the best prediction performance.

Author Response

The authors are grateful for the valuable comments that made it possible to improve the paper.

Here are the responses to the comments.

Point 1: The main innovation and contribution of this research should be clearly clarified in abstract and introduction.

Response 1: In accordance with your recommendation and the recommendations of other reviewers, we added the an additional description of the main innovation to the section Methodology. Lines 99-109

Point 2: Broaden and update literature review on machine learning in data analysis application. E.g. Torsional capacity evaluation of RC beams using an improved bird swarm algorithm optimised 2D convolutional neural network; Nonlinear hysteretic parameter identification using an attention-based long short-term memory network and principal component analysis.

Response 2: The review added with the following links Lines 53-55, 543-546

Point 3: Please add the information about how the authors select the optimal hyperparameters of machine learning models to achieve the best prediction performance.

Response 3: At this stage of the work, we have selected several models that give the best predictions for both speed and accuracy. For most of them, the hyperparameter values ​​are taken equally by default. For other models, the parameters are taken by simple selection. Nevertheless, further research will aim to find an algorithm for selecting the optimal values ​​of hyperparameters since the system's operation does not provide a correction function in real processes. Linew 428-433

Round 2

Reviewer 1 Report

This study considers constructing a time series forecasting model with seasonal variability. Several methods including SARIMA, Holt-Winters Exponential Smoothing, ETS, Facebook Prophet, XGBoost, and Long Short-Term Memory are used. Although I gave a chance to the authors to improve the shortcoming of their manuscript. However, the authors in general ignored my suggestions and hence I am unable to recommend this paper for publication keeping in view the standard of the journal. Some specific reasons for my decision are:

  1. The paper writing quality is quite poor. It needs professional proofreading.
  2. The novelty of the current work is unclear.
  3. Most importantly, the authors used a very short testing period which is not acceptable in the scientific world. The models can perform much better on a certain day and very badly on other days. I ask the authors to use a longer testing period and report the average errors, however, they completely ignored my suggestions. 
  4. I asked the authors to provide proper tables in the paper, however, they have provided the software output. This shows their nonseriousness in their research work.
  5. The paper is not properly structured.

Author Response

Point 1: The paper writing quality is quite poor. It needs professional proofreading.

Response 1: The article has been corrected in accordance with the comments

Point 2: The novelty of the current work is unclear.

Response 2: The novelty was highlighted in the Abstract and Introduction sections, so please reread it.

Point 3: Most importantly, the authors used a very short testing period which is not acceptable in the scientific world. The models can perform much better on a certain day and very badly on other days. I ask the authors to use a longer testing period and report the average errors, however, they completely ignored my suggestions.

Response 3: It seems like the reviewer did not read our response provided earlier. First of all – it was a request from our technical partner to get an algorithm that can use a short dataset for the model training and testing. And the main thing is that we did not always use the same model, and we are using a retrained approach to save accuracy using the short testing and training period.

Point 4: I asked the authors to provide proper tables in the paper, however, they have provided the software output. This shows their nonseriousness in their research work.

Response 4: It’s not a table, and it’s a part of an experiment that confirms the time resources for the model training. That’s why we save the original format. Please note in addition to research and theory, there is practice.

Point 5: The paper is not properly structured.

Response 5: The manuscript edited as recommended by reviewers and follows the MDPI template. The note “The paper is not properly structured” is not very constructive and unclear.

Reviewer 3 Report

Dear Authors,

You performed a lot of improvements.

The list of references still needs to be augmented.

All the best!

Author Response

The authors are grateful for the valuable comments that made it possible to improve the paper. Thank you for your time and all recommendations.

Here are the responses to the comments.

Point 1: The list of references still needs to be augmented.

Response 1: The list of references extended. Lines 45-56, 61-68, 76-77 492-508, 542-547, 550-552

Reviewer 4 Report

The authors have taken into account all my suggestions, improving the quality of their manuscript. I consider this paper publishable in the current form.

Round 3

Reviewer 1 Report

The paper has serious flaws in the experimental section.